# Chemical Composition and Skin-Whitening Activities of *Siegesbeckia glabrescens* Makino Flower Absolute in Melanocytes

**DOI:** 10.3390/plants12233930

**Published:** 2023-11-22

**Authors:** Da Kyoung Lee, Kyung Jong Won, Do Yoon Kim, Yoon Yi Kim, Hwan Myung Lee

**Affiliations:** 1Division of Cosmetic and Biotechnology, College of Life and Health Sciences, Hoseo University, Asan 31499, Republic of Korea; ekrud2877@naver.com (D.K.L.); doyoon@hoseo.edu (D.Y.K.); 20202020@vision.hoseo.edu (Y.Y.K.); 2Korea Essential Oil Resource Research Institute, Hoseo University, Asan 31499, Republic of Korea; 3Department of Physiology and Medical Science, College of Medicine, Konkuk University, Chungju 27478, Republic of Korea; kjwon@kku.ac.kr

**Keywords:** *Siegesbeckia glabrescens* Makino, absolute, skin whitening, melanogenesis, melanocytes, B16BL6 melanoma cells

## Abstract

*Siegesbeckia glabrescens* Makino (SGM) has been traditionally used to treat many disorders, including rheumatoid arthritis, hypertension, and acute hepatitis. However, the biological activities of SGM in skin remain unclear. The present study explored the effects of SGM flower absolute (SGMFAb) on skin-whitening-linked biological activities in B16BL6 cells. SGMFAb was extracted using hexane, and its composition was analyzed through gas chromatography/mass spectrometry analysis. The biological effects of SGMFAb on B16BL6 melanoma cells were detected via WST and BrdU incorporation assays, ELISA, and immunoblotting. SGMFAb contained 14 compounds. In addition, SGMFAb was noncytotoxic, attenuated the serum-induced proliferation of, and inhibited melanin synthesis and tyrosinase activity in α-MSH-exposed B16BL6 cells. SGMFAb also reduced the expressions of MITF (microphthalmia-associated transcription factor), tyrosinase, tyrosinase-related protein (TRP)-1, and TRP-2 in α-MSH-exposed B16BL6 cells. Moreover, SGMFAb downregulated the activation of p38 MAPK, ERK1/2, and JNK in α-MSH-stimulated B16BL6 cells. In addition, SGMFAb reduced the expressions of three melanosome-transport-participating proteins (myosin Va, melanophilin, and Rab27a) in α-MSH-stimulated B16BL6 cells. These results indicate that SGMFAb positively influences skin whitening activities by inhibiting melanogenesis and melanosome-transport-related events in B16BL6 cells, and suggest that SGMFAb is a promising material for developing functional skin whitening agents.

## 1. Introduction

Melanin, a biological pigment produced by melanocytes, protects against the harmful effects of UV radiation and is a major contributor to skin color [1,2]. However, excessive melanin in skin causes dermatological problems and hyperpigmentation disorders including age spots, lentigo, and melasma [1]. In this context, melanin hyperpigmentation can cause cosmetic problems and seriously impact quality of life [3,4]. Therefore, strategies are needed to prevent hyperpigmentation and lighten skin. Melanin is synthesized by melanogenic molecules and stored in melanosomes within melanocytes [5]. Melanosomes containing melanin are transported to the dendritic tips of melanocytes, exit the cells, and are transferred to keratinocytes for distribution throughout skin [5,6]. Therefore, abnormalities in melanin synthesis and melanosome transport within melanocytes are probably associated with melanin hyperpigmentation [5,6].

Melanogenesis is triggered by various stimulating factors, including UV, toxic drugs, or α-melanocyte stimulating hormone (MSH), and involves complex biochemical and enzymatic processes in melanosomes [2]. Melanin synthesis is mediated by several melanogenesis-associated proteins, tyrosinase, tyrosinase-related proteins-1 (TRP-1) and TRP-2, and microphthalmia-associated transcription factor (MITF) [1]. Tyrosinase, TRP-1, and TRP-2 are important enzymes in the multistep reactions that constitute melanogenesis [1]. During melanin biosynthesis, tyrosinase serves as a rate-limiting enzyme that catalyzes the hydroxylation of L-tyrosine to 3,4-dihydroxy-L-phenylalanine (L-DOPA) and the oxidization of L-DOPA to L-DOPA quinone [7]. TRP-1 oxidizes 5,6-dihydroxyindole-2-carboxylic acid (DHICA) to yield 5,6-indolequinone-2-carboxylic acid (IQCA), and TRP-2 catalyzes the conversion of DOPA chrome into DHICA [7]. MITF is a transcription factor that modulates melanogenesis and melanocyte survival and proliferation [2] and regulates the expressions of TRP-1, TRP-2, and tyrosinase during melanogenesis [1]. The transcription factor is regulated by MAPKs (mitogen-activated protein kinases), namely by extracellularly responsive kinase 1/2 (ERK1/2), c-Jun N-terminal kinase (JNK), and p38 MAPK [8]. Thus, MAPKs have been known to act as regulatory signaling pathways implicated in melanogenesis [9]. 

After melanin has been produced, melanosomes are transported from the perinucleus of melanocytes toward their dendritic tips in an actin-determined manner and delivered to keratinocytes [2,5]. The transport of melanin-containing melanosomes within human melanocytes is mediated through the formation of the Rab27a/melanophilin (also known as Slac2-a)/myosin Va (also known as MyoVa) complex, and thus, Rab27a, melanophilin, and myosin Va play critical roles in the regulation of actin-dependent melanosome transport within melanocytes [5]. Rab27a is a member of the small GTPase Rab family and recruits melanophilin and myosin Va to the melanosome membrane, allowing melanosomes to connect to the actin cytoskeleton in the plasma membrane of melanocytes [10]. During the melanosome transport step, melanophilin interacts with melanosome-bound Rab27a and acts as a linker protein between myosin Va and Rab27a during melanosome transport [11]. Myosin Va is an actin-based motor protein that causes melanosomes to be captured in the peripheral subcortical actin network at the tips of dendrites [12]. 

*Siegesbeckia glabrescens* Makino (SGM; family Compositae) is an annual herb distributed in Korea, Japan, and China [13]. SGM has been widely used as a traditional medicinal plant to treat disorders such as rheumatoid arthritis, hypertension, paralysis, and acute hepatitis [14]. Extracts or compounds derived from SGM have many biological activities, including anti-allergic, anti-inflammatory, antibacterial, anti-angiogenic, and anti-cancer effects [14,15,16,17,18]. In addition, it has also been reported that an SGM-derived extract and its constituent compounds have antioxidant and anti-melanogenic effects [19]. However, the skin-whitening-related activities of SGM or its extracts have not been previously investigated. In this study, we investigated the effect of the absolute extracted from SGM flowers (SGMFAb) on skin-whitening-related responses in melanocytes (B16BL6 melanoma cells).

## 2. Results

### 2.1. Chemical Composition of SGMFAb

Gas chromatography/mass spectrometry (GC/MS) identified 14 compounds in SGMFAb (Figure 1 and Table 1). In order of abundance, the compounds present were lauric acid (34.41%), methyl undecanoate (25.65%), α-springene (12.25%), undecanoic acid (7.21%), γ-elemene (6.16%), oxacyclotetradeca-4,11-diyne (3.72%), 2-isopropyl-5-methylanisole (2.85%), tridecanoic acid (2.65%), trans-α-bisabolene (1.42%), and others (Table 1).

### 2.2. Effects of SGMFAb on the Viability and Proliferation of B16BL6 Melanoma Cells

We first assessed whether SGMFAb was toxic to B16BL6 cells by using the WST (water-soluble tetrazolium salt) assay. SGMFAb (0.1–20 μg/mL) had no significant cytotoxic effect at concentrations of 0.1–5 μg/mL and 20 μg/mL, but significantly increased B16BL6 cell viability at a concentration of 10 μg/mL (Figure 2a). Thus, SGMFAb concentrations within the range of 0.1 to 20 μg/mL were used to examine its effects on B16BL6 cells. We next analyzed the effect of SGMFAb on B16BL6 cell proliferation using the BrdU (5-bromo-2′-deoxyuridine) assay. Experiments using SGMFAb concentrations from 0.1 to 20 μg/mL showed that at 10 and 20 μg/mL, SGMFAb significantly reduced FBS (2%)-induced B16BL6 cell proliferation and that this effect was greatest at 20 μg/mL, at which it reduced B16BL6 cell proliferation to 71.86 ± 2.12% of the untreated control (Figure 2b).

### 2.3. Effects of SGMFAb on Melanin Synthesis and Tyrosinase Activity in B16BL6 Melanoma Cells

To determine whether SGMFAb affects melanin synthesis in B16BL6 cells, we measured melanin contents after treating α-MSH (200 nM)-stimulated B16BL6 cells with SGMFAb at 0.1–20 μg/mL. All expression levels are expressed as percentages of 2% FBS-alone-treated controls, unless otherwise stated. An amount of 200 nM of α-MSH markedly increased the level of melanin content in B16BL6 cells to 232.63 ± 0.88%, and this was significantly reduced by SGMFAb (1–20 μg/mL) in a concentration-dependent manner and reached a maximum of 110.66 ± 0.00% at an SGMFAb concentration of 20 μg/mL (Figure 3a). Additionally, we also observed the effect of SGMFAb on tyrosinase activity in melanocytes. Treatment with SGMFAb (5–20 μg/mL) significantly attenuated 200 nM α-MSH-induced tyrosinase activity and this peaked at 126.75 ± 0.60% at 20 μg/mL of SGMFAb (Figure 3b).

### 2.4. Changes in Expression of Melanogenesis Regulatory Molecules Induced by SGMFAb in B16BL6 Melanoma Cells

To determine whether reductions in melanogenesis and tyrosinase activity in SGMFAb-exposed B16BL6 cells are associated with key melanogenic enzymes and their upstream transcription factor, the expression levels of tyrosinase, TRP-1, TRP-2, and MITF were determined through Western blotting. All expression levels are expressed as percentages of 2% FBS-alone-treated controls, unless otherwise stated. α-MSH (200 nM) enhanced tyrosinase expression to 313.43 ± 15.34% in B16BL6 cells (Figure 4a,b). Treatment with SGMFAb (0.1 to 20 μg/mL) significantly reduced the α-MSH-induced expression of tyrosinase in B16BL6 cells at 10 and 20 μg/mL and peaked at 20 μg/mL (70.70 ± 15.34%) (Figure 4a,b). Moreover, 200 nM α-MSH elevated TRP-1 expression to 222.32 ± 24.50%, and this showed a significant reduction through treatment with SGMFAb at 20 μg/mL to 113.43 ± 26.38% (Figure 4a,c). In addition, SGMFAb significantly inhibited the α-MSH-increased TRP-2 expression in B16BL6 cells at concentrations of 5 to 20 μg/mL, which reached a maximum at 20 μg/mL (33.45 ± 5.97%) (Figure 4a,d). Furthermore, SGMFAb at concentrations of 10 and 20 μg/mL also significantly decreased the α-MSH-increased MITF expression in B16BL6 cells, peaking at 20 μg/mL (85.28 ± 12.74%; Figure 4a,e).

### 2.5. Changes in the Phosphorylations of MAPKs in B16BL6 Melanoma Cells Exposed to SGMFAb

To test whether SGMFAb influences upstream signaling molecules associated with melanogenesis, we performed a Western blotting analysis to observe changes in the activations of MAPKs (MITF regulators) and in the activations of regulatory signaling molecules implicated in melanogenesis [8,9]. All expression levels are expressed as percentages of 2% FBS-alone-treated controls, unless otherwise stated. α-MSH (200 nM) elevated ERK1/2 phosphorylation to 314.59 ± 19.51% (Figure 5a,b) in B16BL6 cells in MEM containing FBS (2%). Treatment with SGMFAb at 5 to 20 μg/mL significantly inhibited α-MSH (200 nM)-induced ERK1/2 phosphorylation, and this peaked at 20 μg/mL (118.26 ± 28.73%; Figure 5a,b). In addition, SGMFAb at 10 and 20 μg/mL significantly attenuated α-MSH (200 nM)-induced p38 MAPK (Figure 5a,c) and JNK expressions (Figure 5a,d) in B16BL6 cells exposed to FBS (2%). The maximal phosphorylation levels of p38 MAPK and JNK after treating cells exposed to FBS (2%) containing 20 μg/mL of SGMFAb were 60.67 ± 5.57% (Figure 5c) and 58.12 ± 13.14% (Figure 5d), respectively.

### 2.6. Changes in the Expressions of Melanosome-Transport-Related Proteins by SGMFAb in B16BL6 Melanoma Cells

To test whether SGMFAb affects the expressions of melanosome transport proteins, we performed a Western blotting assay to observe the effects of SGMFAb on myosin Va, melanophilin, and Rab27a expression in α-MSH-treated B16BL6 cells in MEM containing FBS (2%). All expressions are expressed as percentages of 2% FBS-alone-treated controls, unless otherwise stated. α-MSH (200 nM) enhanced the expression of myosin Va to 168.39 ± 8.45% (Figure 6a,b). However, SGMFAb (1–20 μg/mL), in a concentration-dependent manner, significantly attenuated the α-MSH (200 nM)-increased myosin Va expression in B16BL6 cells exposed to FBS (2%) with a maximal effect at 20 μg/mL (47.94 ± 12.77%) (Figure 6a,b). In addition, SGMFAb significantly inhibited the expressions of melanophilin (Figure 6a,c) and Rab27a (Figure 6a,d) induced by α-MSH (200 nM) in cells exposed to FBS (2%) at 5 to 20 μg/mL and at 10 and 20 μg/mL, respectively. The expression levels of melanophilin and Rab27a peaked after treatment with 20 μg/mL of SGMFAb (49.63 ± 12.87% (Figure 6c) and 44.27 ± 7.58% (Figure 6d), respectively).

## 3. Discussion

The controlled regulation of skin pigmentation can protect the skin from diseases related to pigmentation and photoaging and improve cosmetic appearance and quality of life. For these reasons, this has been a long-standing issue in the cosmetic and pharmaceutical industries [20]. Many researchers have attempted to develop agents that improve abnormal pigmentation and whiten skin, but many of the developed products do not produce satisfactory results, because of their cytotoxicities and poor efficacies [21,22]. Thus, there is a need for improved skin whitening agents that are more efficient and safer. Natural products, such as plants and plant extracts, are generally non-toxic and thus are promising materials for developing safe and effective agents for depigmenting or whitening the skin [23]. In the present study, we extracted SGMFAb from flowers of *Siegesbeckia glabrescens* Makino and investigated whether SGMFAb has skin whitening or depigmentation properties using B16BL6 cells. These cells were derived from mouse melanoma and stably synthesize melanin and have been widely used as an in vitro model for testing skin whitening or hyperpigmentation effects [24]. 

Melanin synthesis by melanocytes and/or the distribution of melanin in the epidermal layer are associated with skin pigmentation, and their regulation can lead to the induction of skin whitening or depigmentation [25]. Melanin biosynthesis can be affected by the differentiation and proliferation of melanocytes [26]. For example, Zhang et al. found that reduced melanocyte proliferation inhibited melanogenesis [27]. In the present study, SGMFAb attenuated serum-increased B16BL6 cell proliferation, had no cytotoxic effect, and reduced α-MSH-induced melanin production in these cells. Therefore, our results imply that SGMFAb might have the ability to inhibit melanogenesis in melanocytes by inhibiting cell proliferation or possibly by inhibiting other processes. 

Many researchers have reported that skin hyperpigmentation can be prevented by inhibiting melanogenesis, and thus, whiten skin [28]. Furthermore, tyrosinase, TRP-1, and TRP-2 are enzymes that can directly modulate the melanin biosynthesis pathway [29]. Tyrosinase acts as the rate-limiting enzyme in melanin biosynthesis and plays key roles in the first two steps of this process [7], in hydroxylating L-tyrosine to L-DOPA and oxidizing L-DOPA to L-DOPA quinone [7]. It has been established that the activity and protein levels of tyrosine are closely associated with melanin production [30]; for example, reduced tyrosinase activity and protein expression and treatment with a tyrosinase inhibitor attenuated the increase in melanin production induced by α-MSH in B16BL6 cells [31]. Similarly, we found that SGMFAb treatment reduced α-MSH-induced increases in tyrosinase activity and its expression in B16BL6 cells. In addition, undecanoic acid, one of the 14 components of SGMFAb, has been reported to have anti-tyrosinase activity [32], which implies that an association exists between undecanoic acid and the tyrosinase inhibitory effect of SGMFAb. However, none of the other 13 components are known to be associated with anti-pigmentation responses, including anti-tyrosinase activity. Also, SGMFAb downregulated the expressions of TRP-1 and TRP-2 and the production of melanin in B16BL6 cells exposed to α-MSH. Joo et al. reported that *Ligularia fischeri* ethanol extract inhibited melanin synthesis by reducing α-MSH-induced TRP-1 and TRP-2 expressions in B16F10 cells [33]. Therefore, our findings suggest that SGMFAb may inhibit the activity and expressions of tyrosinase and its two related proteins (TRP-1 and TRP-2) in melanocytes. 

MITF is a key transcription factor that induces the expressions of tyrosinase-related proteins (TRP-1 and TRP-2) and tyrosinase and promotes melanin synthesis [7]. Furthermore, it has been reported that the expressions of tyrosinase, TRP-1, and TRP-2 are inhibited through MITF downregulation and that this response was associated with reduced melanin production in B16F10 cells [34]. Phytochemicals (kirenol and methyl ent-16α,17-dihydroxy-kauran-19-oate), which were derived from SGM, inhibited melanogenesis by inhibiting MITF-mediated TRP-1, TRP-2, and tyrosinase activation in B16F10 cells [19]. These reports imply that MITF is an upstream molecule in the melanin production pathway in B16F10 cells. As mentioned above, we found that SGMFAb reduced the α-MSH-stimulated MITF expression in B16BL6 cells and that it inhibited TRP-1, TRP-2, and tyrosinase expression in B16BL6 cells. Therefore, our observations suggest that SGMFAb downregulates the expressions of melanogenic enzymes (tyrosinase, TRP-1, and TRP-2) by inhibiting MITF expression.

The function of MITF in melanogenesis can be mediated by MAPKs, such as p38 MAPK, JNK, and ERK1/2, which play important roles in the melanin production pathway [35,36]. Interestingly, it was reported that an anticoagulant drug had an anti-melanogenic effect by downregulating p38 MAPK and JNK signaling, upregulating ERK1/2 expression, and thus reducing MITF expression in melanocytes [36]. Also, p38 MAPK and JNK inhibitors suppressed melanogenesis by downregulating MITF and tyrosinase in B16F10 cells [35]. These reports indicate that MITF is a downstream molecule in the MAPK signaling pathway associated with melanogenesis in melanocytes. However, the relationship between ERK1/2 and MITF in the melanogenesis pathway is controversial; for example, ERK1/2 activation induced melanin synthesis by reducing MITF expression in melanocytes [37,38], whereas ERK1/2 activation inhibited melanogenesis by reducing levels of melanogenesis-regulatory proteins, including that of MITF [39]. In the present study, we showed that SGMFAb reduced the expression of MITF and phosphorylations of MAPKs (JNK, p38 MAPK, and ERK1/2) in B16BL6 cells exposed to α-MSH. In addition, SGMFAb also inhibited melanin production and downregulated the expression levels of tyrosinase, TRP-1, and TRP-2 protein in B16BL6 cells exposed to α-MSH. Furthermore, MITF stimulates the expressions of melanogenic proteins and thus promotes melanin synthesis [7], and thus, our findings suggest that the anti-melanogenic activity of SGMFAb is due to MAPK-mediated signaling pathways probably leading to the downregulation of MITF.

After melanin is synthesized in melanosomes within skin melanocytes, melanin-containing mature melanosomes are transported to dendritic tips by actin-based motor proteins along microtubules and actin filaments [40]. Melanosomes are released from cells through exocytosis and transferred to keratinocytes for distribution in the skin [6,40], implying that melanosome transport influences skin pigmentation by moving melanin pigments within melanocytes. Myosin Va, melanophilin, and Rab27a form a tree component complex to mediate actin-dependent melanosome transport within melanocytes [41,42,43,44]. Depletion or defects in the proteins involved in the formation of this complex interfere with melanin transport and cause abnormal melanosome distributions and hypopigmentation [40,45]. Furthermore, these reports suggest that the modulation of molecules responsible for melanosome transport might be used to whiten or depigment skin. It has also been reported that the absence of one of these three proteins causes defects in melanosome transport that lead to the perinuclear aggregation of melanosomes associated with hypopigmentation [5]. 

The downregulation of Rab27a attenuated UVB-induced melanogenesis and melanosome transport in melanocytes [46], and the knockdown of myosin Va isoforms abolished melanosome transport within melanocytes [47]. Furthermore, reduced levels of melanophilin and myosin Va in melanocytes by salicylic acid, a plant-derived compound, inhibited melanosome transport from melanocyte perinuclear areas to dendrite tips and was associated with the inhibition of pigmentation, leading to skin whitening [48]. Moreover, a natural compound reduced the expression levels of myosin Va, melanophilin, and Rab27a in melanocytes and inhibited melanin transport and skin pigmentation in an ex vivo human skin model and in vivo model. Based on these results, the authors suggested the natural compound offered a potential means of targeting skin pigmentation [49]. In the present study, we found that SGMFAb reduced α-MSH-induced increases in myosin Va, melanophilin, and Rab27a levels in B16BL6 cells. Thus, our findings imply that SGMFAb may inhibit melanosome transport from the melanocyte perinuclear region to dendrite tips by inhibiting the expressions of molecular motor proteins (myosin Va, melanophilin, and Rab27a). In addition to the melanin transport inhibitory effect of SGMFAb, we also found that SGMFAb inhibited melanogenesis in melanocytes, which suggests that SGMFAb has bifunctional effects, namely, that it induces skin depigmentation or whitening by inhibiting intracellular melanosome transport and melanogenesis in melanocytes.

## 4. Materials and Methods

### 4.1. Materials

Fetal bovine serum (FBS), penicillin/streptomycin (P/S), and trypsin-ethylenediaminetetraacetic acid were purchased from Hyclone (Logan, UT, USA), and minimum essential medium (MEM) and phosphate-buffered saline (PBS) were obtained from Welgene (Daegu, Korea). Dimethyl sulfoxide (DMSO), α-MSH, L-DOPA, Triton X-100, β-Actin were obtained from MilliporeSigma (St. Louis, MO, USA). Bovine serum albumin was obtained from GenDEPOT (Katy, TX, USA) and the EZ-CyTox kit was obtained from DoGenBio (Seoul, Korea). Antibodies for JNK, phospho JNK, p38 MAPK, phospho p38 MAPK, ERK1/2, phospho ERK 1/2, MITF, myosin Va, rabbit immunoglobulin G, and mouse immunoglobulin G were purchased from Cell Signaling (Beverly, MA, USA). Antibodies for tyrosinase, TRP-1, and TRP-2 were obtained from Abcam (Cambridge, UK). Anti-Rab27a was from Santa Cruz Biotechnology (Dallas, TX, USA), and anti-melanophilin was from Proteintech (Rosemont, IL, USA).

### 4.2. Extraction of Siegesbeckia glabrescens Makino Flower Absolute

SGM flowers were collected on 14 September 2017 from Hanaro Farm (Songji-myeon, Jeollanam-do, Korea, 34°23′00.4″ N 126°33′59.0″ E). The plant and its flowers were authenticated by Jong-Cheol Yang from the Division of Forest Biodiversity and Herbarium, Korea National Arboretum (Pocheon-si, Republic of Korea). A voucher specimen (No. SGM-003) was stored at the Herbarium of the College of Life and Health Science (Hoseo University, Asan-si, Republic of Korea). Absolute was obtained through solvent extraction method, according to previous report [24,50]. Briefly, 2.58 kg of SGM flowers were completely soaked in hexane (Samchun, Pyeongtaek, Republic of Korea) at room temperature (RT) for 1 h. After obtaining the extracts, they were evaporated to remove hexane using a rotary evaporator (EYELA, Tokyo, Japan) at 25 °C under vacuum to produce a dark yellow waxy residue (concrete). The concrete was mixed with ethanol (99.5%; Samchun, Pyeongtaek-si, Republic of Korea) and kept at −20 °C for 12 h, followed by filtering through a sintered funnel. After evaporating the ethanol at 35 °C, a light-yellow anhydrous wax (SGMFAb; 5.6 g, yield 0.217%, *w*/*w*) was obtained. SGMFAb was kept at −80 °C for further experiments.

### 4.3. Analysis of Chemical Compounds of SGMFAb

SGMFAb analysis was conducted by NICEM (the National Instrumentation Center for Environmental Management, Seoul National University, Korea). Identification of its components was carried out using GC/MS analysis using a TRACE 1310 GC unit coupled to an ISQ LT single-quadrupole mass spectrometer (Thermo Scientific, Waltham, MA, USA), as described previously [24]. Briefly, derivatized samples were subjected to separation on a DB-5MS column (60 m × 0.25 mm, 0.25 μm; Agilent Technologies, Santa Clara, CA, USA) at a constant flow rate of 1 mL/min using the following program: 50 °C for 5 min, 50 to 65 °C at 10 °C/min, 65 to 210 °C at 5 °C/min, 210 to 310 °C at 20 °C/min, and 310 °C for 10 min. Mass spectra were acquired in the range of *m*/*z* 35 to mz 550 at rate of 0.2 scans/s. Transfer line and ion source temperatures were 300 °C and 270 °C, respectively. The identification of detected compounds was conducted by comparing mass spectra and retention indices (RIs) with reference standards in the NIST/NIH/EPA mass spectral library (NIST 11, version 2.0 g) and by matching retention times and spectra with those of commercially available standards. A solution of C_7_–C_30_ *n*-alkanes as standards was used to calculate RIs.

### 4.4. Cell Culture

B16BL6 murine melanoma cell line was purchased from the Korean Cell Line Bank (KCLB, Seoul, Korea). B16BL6 cell culture was conducted in MEM containing 10% (*v*/*v*) FBS and 1% (*v*/*v*) P/S in a humidified 95% air/5% CO_2_ atmosphere at 37 °C in an incubator. Cells were incubated until 70–80% confluence for experiments. 

### 4.5. Cell Viability Assay

B16BL6 cell viabilities were analyzed through WST assay using the EZ-CyTox kit (DoGenBio, Geumcheon-gu, Seoul, Korea) [24]. Briefly, cells were plated into 96-well microtiter plates at a density of 2 × 10^3^ cells/well. After 12 h incubation, different concentrations of SGMFAb in MEM with 2% FBS and 0.2% DMSO were added to each well. The plate was incubated for 48 h and then loaded with EZ-CyTox reagent (30 μL/well) for 30 min at 37 °C. Absorbance at 450 nm was determined using a multi-well plate reader (Synergy 2; BioTek Instruments, Winooski, VT, USA).

### 4.6. Proliferation Assay

B16BL6 cell proliferation was assayed using a BrdU proliferation ELISA kit (Roche, Indianapolis, IN, USA) [24,51]. Briefly, 2 × 10^3^ cells were seeded in each well of 96-well plates and cultured for 12 h. Different concentrations of SGMFAb in MEM containing 0.2% DMSO with or without 2% FBS were added to each well, followed by incubation for 36 h. The cells were labeled in BrdU-labeling solution (10 μM) for 12 h at 37 °C, and the culture medium was then removed. After fixing and denaturing the DNA with FixDenat solution from the BrdU kit for 30 min at RT, cells were added with peroxidase-labeled anti-BrdU monoclonal antibody and incubated at RT for 90 min. BrdU-antibody complexes were detected using a luminometer (Synergy 2; BioTek Instruments).

### 4.7. Melanin Content Assay

Melanin content assay was performed according to the method reported previously [24,52]. In brief, 1 × 10^5^ B16BL6 cells were cultured in each dish in 60 mm culture dishes for 12 h. Cells were then incubated in MEM (containing 2% FBS) in the presence or absence of various concentrations of SGMFAb with or without 200 nM α-MSH for 48 h at 37 °C. Cells were washed with PBS and lysed with 0.1 M sodium phosphate buffer (pH 6.8) with 0.2 mM phenylmethylsulfonyl fluoride and 1% Triton X-100, followed by centrifugation (10,000× *g*, 15 min). The obtained cell pellets were lysed in 1 N NaOH solution with 10% DMSO and incubated at 80 °C for 1 h. Absorbances were recorded at 405 nm using an ELISA reader (Synergy 2; BioTek Instruments).

### 4.8. Tyrosinase Activity Assays

Tyrosinase activity was analyzed based on the dopachrome method using L-DOPA as substrate, as reported previously [24,51]. In brief, B16BL6 cells (1 × 10^5^ cells/dish) were cultured in 60 mm culture dishes for 12 h and then incubated in MEM (with 2% FBS) in the presence or absence of SGMFAb at different concentrations with or without α-MSH (200 nM) for 48 h at 37 °C. Cells were then lysed and centrifuged as in the melanin content assay described above. A mixture of 60 μL of supernatants and 140 μL of 2 mg/mL L-DOPA solution were then added to each well of a 96-well plate. After incubation at 37 °C for 60 min, the absorbance was determined at 490 nm using an ELISA reader (Synergy 2; BioTek Instruments).

### 4.9. Western Blot Analysis

Western blot analysis was carried out as in a previous report [24,51]. Cells were lysed using RIPA (radioimmunoprecipitation assay) buffer (Cell Signaling) and centrifuged (17,000× *g*, 15 min, 4 °C). Total protein concentrations in supernatants obtained were measured using DC (detergent compatible) protein assay kit (Bio-Rad Laboratories, Hercules, CA, USA). Proteins (10–100 μg/lane) were separated through 8–12% sodium dodecyl sulfate polyacrylamide gel electrophoresis and electrophoretically transferred onto polyvinylidene fluoride membranes (PVDF) at 4 °C. The PVDF membranes were incubated with blocking solution (3% skim milk) at RT for 2 h. After washing with PBS containing 0.05% Tween-20, membranes were loaded with primary antibodies (diluted 1:1000 to 5000) and then incubated with horseradish-peroxidase-conjugated secondary antibody at RT for 1 h. Protein band detection was performed using a chemiluminescence substrate and a chemiluminescence imaging system (LuminoGraph, ATTO, Tokyo, Japan).

### 4.10. Statistical Analysis

Statistical analyses of the results were conducted using GraphPad Prism version 5.0 (GraphPad Software, Inc., La Jolla, CA, USA). All experimental data are presented as means ± standard errors of means (SEMs). Student’s *t*-test was performed to assess the significance of the differences between pairs of groups. A one-way ANOVA (analysis of variance) with Tukey’s post hoc test was performed to analyze the significances of differences between multiple groups. Differences with *p* values of less than 0.05 were accepted as statistically significant.

## 5. Conclusions

In this study, 14 compounds were identified in SGMFAb. Furthermore, SGMFAb inhibited serum-induced B16BL6 melanoma cell proliferation and attenuated α-MSH-induced melanin production and tyrosinase activity in B16BL6 melanoma cells. SGMFAb also reduced the expressions of melanogenic proteins (MITF, tyrosinase, TRP-1, and TRP-2) in α-MSH-stimulated B16BL6 cells. In addition, SGMFAb downregulated the phosphorylations of MAPKs (ERK1/2, p38 MAPK, and JNK) in α-MSH-stimulated B16BL6 cells and attenuated α-MSH-induced increases in the expressions of three melanosome-transport-related proteins (myosin Va, melanophilin, and Rab27a). These findings indicate that SGMFAb may promote skin whitening by suppressing melanogenesis and/or melanosome transport responses in B16BL6 cells. Therefore, our findings suggest that SGMFAb has potential use for the development of a natural skin whitening or depigmenting agent. Further studies are needed to determine which components of SGMFAb inhibit bioactivities related to melanogenesis and melanosome transport in melanocytes. In addition, the application of SGMFAb or its derived compounds as materials for human skin whitening and depigmenting agents may require further studies to verify its anti-melanogenic effects using healthy normal melanocytes and in vivo models. 

## Figures and Tables

**Figure 1 plants-12-03930-f001:**
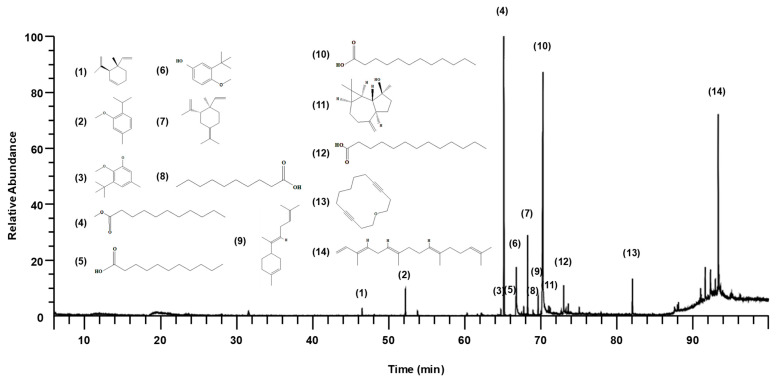
GC/MS total ion chromatogram of *Siegesbeckia glabrescens* Makino flower absolute. Numbers in brackets represent the compound numbers of the 14 compounds in Table 1. Compound numbers and structures are shown.

**Figure 2 plants-12-03930-f002:**
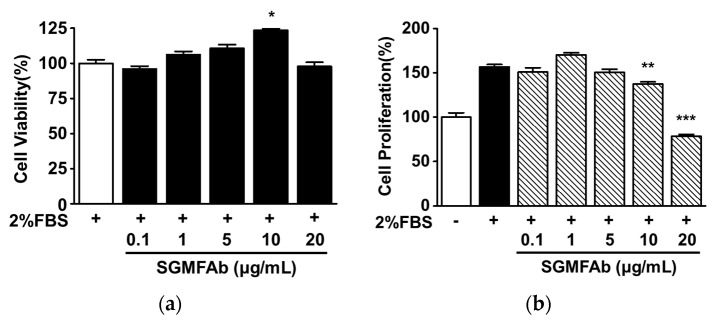
Effects of *Siegesbeckia glabrescens* Makino flower absolute on B16BL6 cell viability and proliferation. (**a**) B16BL6 cell viability. Cells were treated with or without *Siegesbeckia glabrescens* Makino flower absolute (SGMFAb; 0.1–20 μg/mL) in the presence of 2% FBS for 48 h, and then cell viability was quantified using the WST assay (*n* = 3). (**b**) B16BL6 cell proliferation. Cells were incubated with or without *Siegesbeckia glabrescens* Makino flower absolute (SGMFAb; 0.1–20 μg/mL) in MEM in the presence or absence of 2% FBS for 36 h, and the BrdU assay was conducted to test cell proliferation (*n* = 3). Cell viabilities (**a**) and proliferation (**b**) are expressed as percentages of 2% FBS-alone-treated and untreated controls, respectively. Results are presented as means ± S.E.Ms. * *p* < 0.05, ** *p* < 0.01, and *** *p* < 0.001 vs. 2% FBS-alone-treated controls.

**Figure 3 plants-12-03930-f003:**
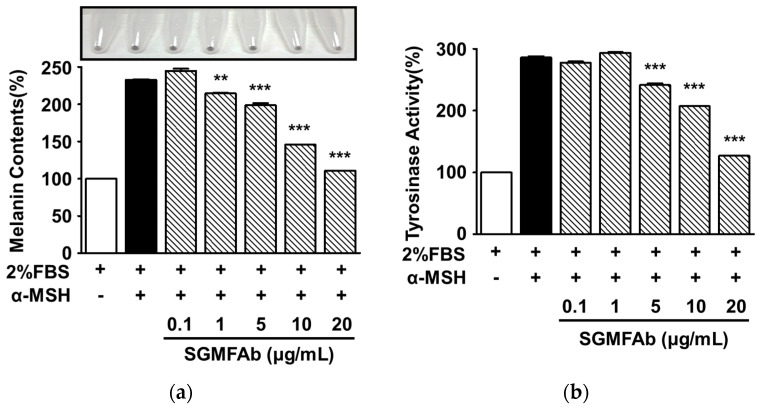
Effects of *Siegesbeckia glabrescens* Makino flower absolute on melanin synthesis and tyrosinase activity in B16BL6 cells treated with α-MSH. B16BL6 cells were incubated at 37 °C for 48 h with or without *Siegesbeckia glabrescens* Makino flower absolute (SGMFAb at 0.1–20 μg/mL in MEM with 2% FBS) in the presence or absence of 200 nM α-MSH. Melanin contents ((**a**); *n* = 3) and tyrosinase activities ((**b**); *n* = 3) were analyzed as described in Section 4. The top picture in panel (**a**) indicates representative result. α-MSH represents a positive control. Data are shown as percentages of levels in FBS (2%)-alone-treated controls as means ± SEMs. ** *p* < 0.01 and *** *p* < 0.001 vs. cells treated with α-MSH alone in the presence of FBS (2%). α-MSH, α-melanocyte-stimulating hormone.

**Figure 4 plants-12-03930-f004:**
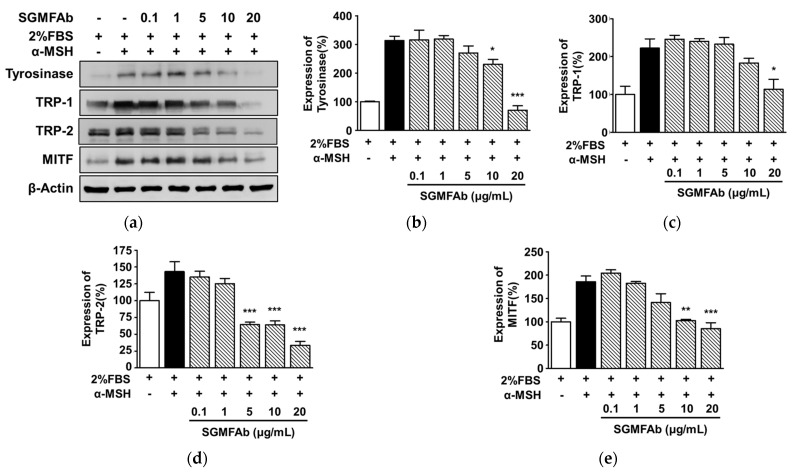
Effect of *Siegesbeckia glabrescens* Makino flower absolute on the expression of melanogenesis-related proteins in B16BL6 cells. (**a**) Representative images. B16BL6 cells were treated with or without *Siegesbeckia glabrescens* Makino flower absolute (SGMFAb at 0.1–20 μg/mL in MEM with 2% FBS) in the presence or absence of 200 nM α-MSH for 24 h. Cell lysates were immunoblotted with the indicated antibodies as described in Section 4. (**b**–**e**) Relative expression levels of tyrosinase ((**b**); *n* = 3), tyrosinase-related protein-1 (TRP-1) ((**c**); *n* = 3), tyrosinase-related protein-1 (TRP-2) ((**d**); *n* = 3), and microphthalmia-associated transcription factor (MITF) ((**e**); *n* = 3). α-MSH indicates a positive control. Expressions are shown as percentages of levels in 2% FBS-alone-treated controls. Data are shown as means ± SEMs. * *p* < 0.05, ** *p* < 0.01, and *** *p* < 0.001 vs. cells treated with α-MSH alone in the presence of 2% FBS.

**Figure 5 plants-12-03930-f005:**
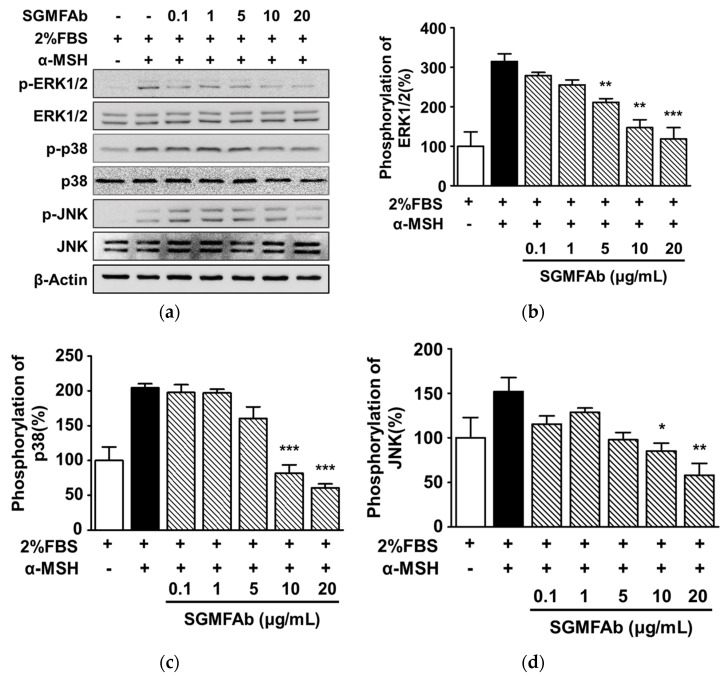
Effect of *Siegesbeckia glabrescens* Makino flower absolute on the phosphorylations of MAPKs in B16BL6 cells. (**a**) Representative images. B16BL6 cells were treated with or without *Siegesbeckia glabrescens* Makino flower absolute (SGMFAb at 0.1–20 μg/mL in MEM with 2% FBS) in the presence or absence of 200 nM α-MSH for 5 min. Cell lysates were subjected to Western blotting with indicated antibodies as described in Section 4. (**b**–**d**) Statistical results for phosphorylated ERK1/2 (**b**), p38 MAPK (**c**), and JNK levels (**d**) obtained from panel (**a**). α-MSH indicates a positive control. The phosphorylation levels of kinase are presented as percentages of levels in 2% FBS-alone-treated controls. Data are shown as means ± SEMs (*n* = 3 for each protein). * *p* < 0.05, ** *p* < 0.01, and *** *p* < 0.001 vs. cells exposed to α-MSH alone in the presence of 2% FBS. P-ERK1/2, phosphorylated ERK1/2; p-JNK, phosphorylated JNK; p-p38, phosphorylated p38 MAPK.

**Figure 6 plants-12-03930-f006:**
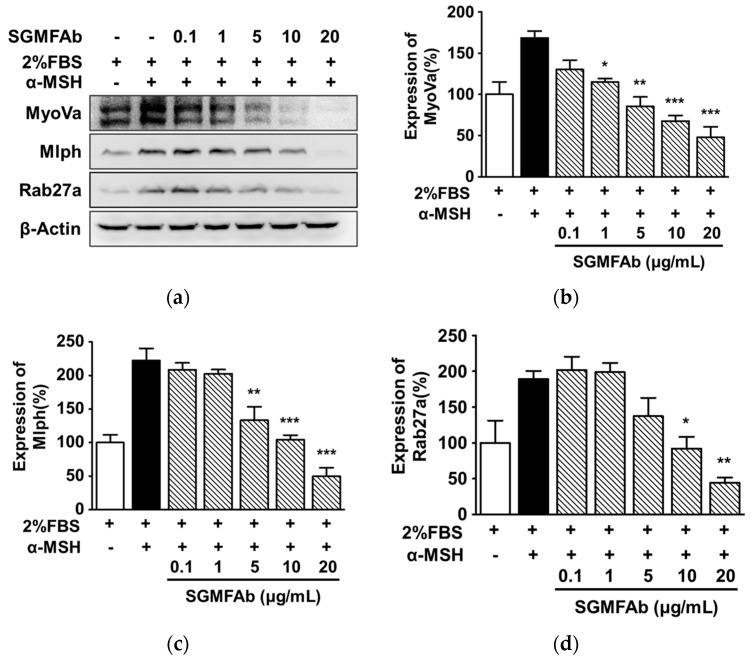
Effect of SGMFAb on the expressions of melanosome transport proteins in B16BL6 cells. (**a**) Representative images. B16BL6 cells were incubated in the presence or absence of *Siegesbeckia glabrescens* Makino flower absolute (SGMFAb 0.1–20 μg/mL in MEM containing 2% FBS) with or without 200 nM of α-MSH for 24 h. Cell lysates were immunoblotted with indicated antibodies. (**b**–**d**) Statistical results for myosine Va (MyoVa; (**b**)), melanophillin (Mlph; (**c**)), and Rab27a (**d**) expression levels obtained from panel (**a**). α-MSH indicates a positive control. The phosphorylation levels of kinases are shown as percentages of levels in controls treated with 2% FBS alone. Results are expressed as means ± SEMs (*n* = 3 for each protein). * *p* < 0.05, ** *p* < 0.01, and *** *p* < 0.001 vs. cells treated with α-MSH alone in the presence of 2% FBS.

**Table 1 plants-12-03930-t001:** Relative compositions of the 14 identified compounds in *Siegesbeckia glabrescens* Makino flower absolute.

No	Component Name	RT ^1^	RI ^2^		Area (%)	CAS No.
			Observed	Literature		
1	Geijerene	46.48	1144	1143	0.73	6902-73-4
2	2-Isopropyl-5-methylanisole	52.17	1227	1227	2.85	1076-56-8
3	2-Methoxy-3-(tert-butyl)-5-methylphenol	64.72	1417	-	0.60	NA
4	Methyl undecanoate	65.12	1426	1427	25.65	1731-86-8
5	Undecanoic acid	66.76	1467	1466	7.21	112-37-4
6	TBMP	67.75	1491	1490	0.78	88-32-4
7	γ-Elemene	68.24	1503	1482	6.16	29873-99-2
8	Capric acid	68.96	1526	1404	0.87	334-48-5
9	trans-α-Bisabolene	69.62	1547	1547	1.42	25532-79-0
10	Lauric acid	70.25	1566	1566	34.41	143-07-7
11	Spathulenol	71.00	1590	1590	0.71	6750-60-3
12	Tridecanoic acid	72.97	1661	1662	2.65	638-53-9
13	Oxacyclotetradeca-4,11-diyne	82.01	1755	1639	3.72	6568-32-7
14	α-springene	93.29	2561	1781	12.25	77898-97-6
Total Identified (%)					100.00	

^1^ RT: retention time, ^2^ RI: retention indices. RTs and RIs were determined using DB-5MS.

## Data Availability

Data are contained within the article.

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
