# Peer review of "Chemical Composition and Skin-Whitening Activities of Siegesbeckia glabrescens Makino Flower Absolute in Melanocytes"

_plants, 2023, doi:10.3390/plants12233930_

Round 1

Reviewer 1 Report

Comments and Suggestions for Authors

This study examined chemical composition and skin-whitening activities of hexane extract of Siegesbeckia glabrescens Makino flower (SGMFAb). The authors identified 14 compounds in the SGMFAb and showed that SGMFAb inhibited melanin synthesis and tyrosinase activity in MSH-exposed B16 melanoma cells. SGMFAb also reduced the expressions of MITF, tyrosinase, and tyrosinase-related proteins. They also found that SGMFAb downregulated the phosphorylations of ERK1/2, p38 MAPK, and JNK in B16 cells and reduced the expressions of three melanosome transport proteins. These findings are good enough to indicate that SGMFAb may promote skin whitening by suppressing melanogenesis and/or melanosome transport in melanocytes. The study appears technically found. However, the reviewer suggests two points to improve this manuscript before acceptance. 

Point 1: SGMFAb contains 14 compounds. Are there any compounds among them that are reported to exhibit the above-mentioned activities? This point is briefly described in Introduction L79-82. However, based on the identification of the 14 compounds, the authors should discuss this point more in Discussion. 

Point 2: There are some typographical errors.

a)     L37: melisma should be melasma.

b)     L49: tyrosine should be tyrosinase

c)     L54: indole-2-carboxylic acid should be indolequinone-2-carboxylic acid.

d)     L72: during should not be in bold.

e)     L110: 71.86 +/- 17.81. “17.81” is not consistent with Figure 2b.

f)      L126: con-centration should be concentration.

g)     L143: associated key should be associated with key.

Author Response

Reviewer 1: Comments and Suggestions for Authors

This study examined chemical composition and skin-whitening activities of hexane extract of Siegesbeckia glabrescens Makino flower (SGMFAb). The authors identified 14 compounds in the SGMFAb and showed that SGMFAb inhibited melanin synthesis and tyrosinase activity in MSH-exposed B16 melanoma cells. SGMFAb also reduced the expressions of MITF, tyrosinase, and tyrosinase-related proteins. They also found that SGMFAb downregulated the phosphorylations of ERK1/2, p38 MAPK, and JNK in B16 cells and reduced the expressions of three melanosome transport proteins. These findings are good enough to indicate that SGMFAb may promote skin whitening by suppressing melanogenesis and/or melanosome transport in melanocytes. The study appears technically found. However, the reviewer suggests two points to improve this manuscript before acceptance.

Comment(C) 1. Point 1: SGMFAb contains 14 compounds. Are there any compounds among them that are reported to exhibit the above-mentioned activities? This point is briefly described in Introduction L79-82. However, based on the identification of the 14 compounds, the authors should discuss this point more in Discussion.

(Response) When we initially submitted the manuscript, we discussed the points about the 14 compounds commented by the reviewer in the Discussion section (L247-251 and L261-263, P9/15; revised manuscript) and Conclusion section (L436-437, P12/15; revised manuscript).  

C2. Point 2: There are some typographical errors.

 C2-1. a) L37: melisma should be melasma.

 (Response) We revised it.

 C2-2. b) L49: tyrosine should be tyrosinase

 (Response) We revised the error.

 C2-3. c) L54: indole-2-carboxylic acid should be indolequinone-2-carboxylic acid

 (Response) We revised it.

 C2-4. d) L72: during should not be in bold

  (Response) We revised the error. Thanks.

 C2-5. e) L110: 71.86 +/- 17.81. “17.81” is not consistent with Figure 2b

  (Response) Based on the reviewer’s comment, we double-checked the data values described in the Results section along with the graphs in all figures, and found that the error bar of graphs created with SEM values were written as the SD values in the Results section. Therefore, descriptions written in SD values, including Figure 2b, in the Results section have been changed to SEM values to match corresponding graphs. We really appreciate the reviewer’s kind and thoughtful comments. (P4/15-P7/15)

Reviewer 2 Report

Comments and Suggestions for Authors

Dear Authors!

Your manuscript needs minor revisions given below.

On page 2, line 49 replace "Tyrosine" with Tyrosinase.

On page 2, line 72 "during" is written in bold but there is no need for that. Correct it please.

On page 4, line 126, change "con-centration-dependent" in "concentration-dependent".

In the Discussion on page 9, line 248 you wrote about survey of literature but no references are given. Please add them.

On page 9, line 253 you wrote about a plant extract. Please name the plant of which you are writting.

On page 10, line 304, again name the plant-derived compound.

Finally, in the Conclusions (or in Discussion) you should write something about further studies on normal melanocytes since your research work was done on melanoma cell line. Maligant melaoncytes differ from normal melanocytes in many ways, maybe the effect of SGMF Ab would be different on normal melanocytes. Skin whitening and depigmentation are treatments only for healthy individuals therefore compounds with such effect should be tested on normal, healthy cells.

Kind regards!

Author Response

Your manuscript needs minor revisions given below.

Comment(C) 1. On page 2, line 49 replace "Tyrosine" with Tyrosinase.

(Response) We did that.

C2. On page 2, line 72 "during" is written in bold but there is no need for that. Correct it please.

(Response) We did that.

C3. On page 4, line 126, change "con-centration-dependent" in "concentration-dependent".

(Response) We did that.

C4. In the Discussion on page 9, line 248 you wrote about survey of literature but no references are given. Please add them.

(Response) We revised it.

C5. On page 9, line 253 you wrote about a plant extract. Please name the plant of which you are writing.

(Response) We revised it.

C6. On page 10, line 304, again name the plant-derived compound.

(Response) Thanks. We revised it.

C2. Finally, in the Conclusions (or in Discussion) you should write something about further studies on normal melanocytes since your research work was done on melanoma cell line. Maligant melaoncytes differ from normal melanocytes in many ways, maybe the effect of SGMF Ab would be different on normal melanocytes. Skin whitening and depigmentation are treatments only for healthy individuals therefore compounds with such effect should be tested on normal, healthy cells.

(Response) As the reviewer knows, B16BL6 melanoma cells have been widely used in in vitro studies of anti-melanogenesis-related responses. This led us to use B16BL6 melanoma cells in the present study. On the other hand, we agree with the reviewer's comments. To apply SGMFAb to human skin whitening and depigmentation, further studies will be needed to investigate the effects of SGMFAb on anti-melanogenesis-related responses using healthy normal melanocytes. We thank the reviewer for constructive comments on our study. We described related contents based on the reviewer’s comments in revised manuscript. (Line 437-440, P12/15).
